# Current Frequency of Mandibular Condylar Process Fractures

**DOI:** 10.3390/jcm12041394

**Published:** 2023-02-09

**Authors:** Marcin Kozakiewicz, Agata Walczyk

**Affiliations:** Department of Maxillofacial Surgery, Medical University of Lodz, 90–549 Lodz, Poland

**Keywords:** mandible, condyle, condylar head, mandible head, fracture, classification, epidemiology, human, surgical treatment, ORIF

## Abstract

There are significant discrepancies in the reported prevalence of condylar process fractures among all mandibular fractures (16.5–56%) in the available literature. In addition, it seems that the actual number of difficult-to-treat fractures of the mandibular head is unknown. The purpose of this study is to present the current incidence of the different types of mandibular process fractures with a special focus on mandibular head fractures. The medical records of 386 patients with single or multiple mandibular fractures were reviewed. Of the fractures found, 58% were body fractures, 32% were angle fractures, 7% were ramus fractures, 2% were coronoid process fractures, and 45% were condylar process fractures. The most common fracture of the condylar process was a basal fracture (54% of condylar fractures), and the second most common fracture was a fracture of the mandibular head (34% of condylar process fractures). Further, 16% of patients had low-neck fractures, and 16% had high-neck fractures. Of the patients with head fractures, 8% had a type A fracture, 34% had a type B fracture, and 73% had a type C fracture. A total of 89.6% of the patients were surgically treated with ORIF. Mandibular head fractures are not as rare as previously thought. Head fractures occur twice as often in the pediatric population than in adults. A mandibular fracture is most likely related to a mandible head fracture. Such evidence can guide the diagnostic procedure in the future.

## 1. Introduction

An increasing number of people in Poland, the Czech Republic, Germany, Sweden and Finland are sustaining fractures to the facial part of the skull, including fractures of the mandibular condylar process [1]. Mandibular condylar process fractures are a major concern. The main focus of this research is the frequency of these types of fractures, specifically the current overall incidence and the predicted incidence in patients with condylar process fractures in hospital emergency rooms. It is a fairly common fracture that has been debated for decades [2,3,4,5,6,7,8]. It concerns classification [9,10,11,12,13,14], as well as treatment techniques [8,15,16,17,18,19,20]. Treatments have evolved over time and have been debated among authorities. Originally, closed treatment in a completely conservative version or with closed reduction and maxillomandibular immobilization was widely recommended. Later, as closed treatment had poor functional results, open reduction and immobilization with skeletal anchorage became the preferred choice. Subsequently, open reduction and rigid fixation (ORIF) were attempted more often. It seems that ORIF, as the preferred choice for treating fractures of the mandibular condylar process, influences the reported epidemiology of these fractures. Treatment has been mixed with diagnosis. In daily clinical work, the diagnosis is made only after the fracture is fixed, and then surgeons suggest the method of osteosynthesis for classifying the fracture. This obscures information about the actual incidence of mandibular condylar process fractures.

The frequency, which ranges from 25% to 55% as described by different authors, is without consensus. At this point, it is worth noting a 1970s publication by Lindahl [9], who recognized the incidence and variation of mandibular head fractures as early as the 1960s. Being an employee of the Department of Oral Radiology, he prudently recognized the inadequacies of radiological imaging and described it as an “insufficient radiographic examination”, and in a few cases, “some radiographs were missing”. He based his observations solely on plain radiography supported by orthopantomograms. He excluded as many as 18 of the 141 cases due to deficiencies in radiological material. Lindahl attempted to divide the mandibular head fractures that occur in patients. He distinguished between horizontal (*h*), vertical (*v*), and compression fractures (*c*). Today, one would classify these *h* fractures as high neck fractures. Later, the author himself also put forward another belief, arguing that they cannot be classified as fractures of the mandibular head because they do not involve the condylar cartilage. In contrast, one would now distinguish such fractures from comminuted fractures (these are *c* fractures, according to Lindahl). Thus, from his group, *v* fractures (i.e., currently known as A, B, and C) account for 18% of the described fractures of the condylar process of the mandible. The incidence of a mandibular condylar process fracture has been reported to vary [9]. The above data were cited to indicate another challenge facing current maxillofacial surgeons, i.e., the widespread availability of highly accurate image acquisition devices. Undoubtedly, this is a factor that improves the efficiency of assessing the incidence of mandibular condylar process fractures, especially those of the mandibular head.

The purpose of this study is to present the current incidence of the different types of mandibular process fractures with a special focus on mandibular head fractures.

## 2. Materials and Methods

The materials consisted of the medical database of the CliniNet program (www.cgm.com, accessed on 1 October 2022) at the Department of Maxillofacial Surgery, Medical University of Lodz, Poland, and the results of computer tomography scans evaluated in RadiAnt (www.radiantviewer.com, accessed on 1 October 2022), which are available in this department. The keywords for selecting patients were mandibular fracture and mandibular condylar process fracture. The research covered the period from 1 January 2020 to 1 October 2022, which was 33 months of continuous operation of the hospital department. Patients were cared for around the clock because the hospital ward provides 24-h medical care and serves a region with a population of approximately 2.5 million. In addition, statistics on the availability of treatment should be provided. There are 39 maxillofacial surgery departments/subdepartments in Poland (38.3 million in 2020, 38.1 million in 2021, and 41.5 million residents, including 3.2 million citizens of Ukraine, in 2022 [21]), of which only three provide surgical treatment for mandibular head fractures. All residents of the country have the option of hospital treatment, and 35.8 million people have health insurance [22].

Fractures of the mandibular body, angle, and ramus, as well as fractures of the base and neck (low neck and high neck) of the mandibular condyle according to Kozakiewicz classification [14] and three types of mandibular head fractures (A, B, and C according to Neff [10,11]), were counted in the archival material of the Department of Maxillofacial Surgery, Medical University of Lodz, Poland, analyzed, and subjected to radiological verification. In addition, data were collected on sex, age, place of residence, and cause of injury.

To interpret the epidemiological results, the prevalence of mandibular head fractures was additionally checked. The amount of medical-scientific research interest in the PubMed database (www.pubmed.ncbi.nlm.nih.gov/advanced/, accessed on 31 December 2022) related to the topic of this publication was checked. Studies related to mandibular head fractures (or intracapsular fractures) and studies related to the surgical treatment of such fractures were searched for. The search query was ((((((mandible head fracture)) OR ((mandible intracapsular fracture)))) AND (((((fixation)) OR ((osteosynthesis))) OR ((surg)))))) AND (1982:2022[pdat])).

Statistical analysis was performed in Statgraphics Centurion 18 (Statgraphics Technologies Inc., The Plains City, VA, USA). A *p* value of less than 0.05 was considered statistically significant.

## 3. Results

A total of 386 patients with mandibular fractures were included in the 3-year investigation. The average age of the treated patients was 34.5 ± 15 years old. There were 68 females (40 ± 19 years old) and 318 males (33 ± 14 years old). Eighty-seven mandibular fractures were registered in the first year of investigation (COVID-19 pandemic year 2020), 150 cases during the second year (2021), and 149 cases during the last year of this study (2022).

### 3.1. Epidemiology

Eighty percent of patients were urban residents, and 20% were rural people. The most common cause of mandibular fracture was assault (51%), followed by accidental falling from one’s own height (26%), bicycle or scooter accident (11%), fall from height (just over 4%), car accident (just under 4%), sports accidents (2%), workplace accidents (1%), falls from stairs (0.26%), and motorcycle accidents (0.26%). See Figure 1 below.

The analysis of the incidence of mandibular fractures in females (18% of subjects in the study group) and males (82% of subjects) (Figure 2) revealed that there were significantly more fractures due to falls (*p* < 0.001) in females, and there were significantly more fractures due to assaults (*p* < 0.001) in males. In the collected material, it can be noted that a fracture of the condylar process of the mandible occurs significantly more often than a fracture in other parts of the mandible when the cause of injury is a bicycle or scooter accident (*p* < 0.001). Detailed epidemiological results are shown in Table 1 and Figure 3. The causes of injury clearly depend on the age and sex of the patients. The elderly suffer fractures to their heads as a result of assaults and falls, while children suffer injuries from bicycle and scooter accidents. In the presented traumatological material, there is an overwhelming majority of men, but the same main causes of injury apply to both sexes, i.e., assaults with beatings and falls.

In the pediatric population, significantly fewer beatings and falls were found, while there were significantly more scooter accidents. This is the most common cause of mandibular fractures in children.

It was noted that assaults and falls are common causes of mandibular angle fractures (*p* < 0.001). High-neck fractures are most commonly caused by falls (*p* < 0.01), and type B and C mandibular head fractures are most commonly caused by falls and road traffic accidents (*p* < 0.001).

It has been noted that falls and vehicle traffic accidents are more likely to cause condylar process fractures (*p* < 0.001) than other mandibular fractures. Although condylar fractures often occur as a result of violence (assaults), beatings are a frequent cause of fractures in areas of the mandible other than the condylar process. Although fractures of the condylar process of the mandible in children (69%) are more common than other fractures of the lower jaw (in contrast to adults, 44%), this difference is not statistically confirmed (*χ^2^* test of independence, *p* = 0.055). Head fractures occurred in older patients (average age: 39.5 ± 17.7 y.o.). The group without fractures of the mandibular head was younger (*p* < 0.05)(33.6 ± 14.2 y.o.). After separating the pediatric population, no such relationship could be observed. On the other hand, examination of mandibular head fractures among adults (15% of all mandibular fractures) and children (31% of all mandibular fractures) shows that mandible head fractures occur more often in children.

### 3.2. Frequency of Fractures

In the group of 386 traumatology patients, 175 patients had a condylar fracture, which represents 45% of all mandibular fractures registered (Figure 4 and Table 2).

Fractures of the condylar process occurred as isolated incidents in 17% of mandibular fractures and as a part of multiple mandibular fractures in 63%. Therefore, if the bilateral fracture of the condylar processes is considered to be an isolated fracture (3%), then all isolated condylar fractures among all mandibular fractures reach the level of 20%. Bilateral condylar fractures occurred among all fractures of the mandible in only 12% of cases, and 38% were unilateral condylar fractures among all fractures of the mandible. Among mandibular condylar process fractures, the ratio of unilateral to bilateral fractures is 75:25.

Among mandibular condylar process fractures, the majority are fractures of the base (54%), but the second most common is a fracture of the mandibular head (34%), and the rarest is a fracture of the neck (16%). In each case where a mandibular head fracture was identified, ORIF with the use of headless compression screws was considered. Finally, 55 fractures out of these 59 were treated surgically.

Nine cases of fracture of the coronoid process were reported (Figure 4 and Figure 5). It was never an isolated fracture. It coexisted with fractures of the mandibular body (5 of 9), mandibular rami (2 of 9), the base of the condylar process (5 of 9), type B mandibular head (1 of 9), and type A mandibular head (1 of 9).

### 3.3. Diagnosis and Its Relations

There are also frequent associations of fracture locations with each other that are statistically significant (Table 3). It is important to emphasize the very often observed common occurrence of a body and/or angle fracture with fractures of the mandibular head (*p* < 0.001). The fracture of the angle of the mandible itself is statistically associated with the occurrence of a fracture of the condylar base (*p* < 0.001) and the fracture of the neck of the mandible on the opposite side (*p* < 0.05). The same applies to the joint occurrence of ramus fractures with the base of the condylar process (*p* < 0.05) and mandibular head fractures (*p* < 0.05). The appearance of bilateral condylar fractures was a combination of fractures of the head of the mandible (*p* < 0.001), the head of the mandible with the neck on the opposite side (*p* < 0.05), or the head of the mandible with the base of the condyle (*p* < 0.05).

Patient sex was not associated with fractures of the body, ramus, coronoid process, base of the condyle, or type A, B, or C head fractures. Fractures of the angle of the mandible were more common in males, contrary to low-neck and high-neck fractures, which were more common in females. Place of residence was not associated with fractures of the body, ramus, coronoid process, base of the condyle, neck of the mandible at any level, or type A, B, or C head fractures. Only mandibular angle fractures occurred more often in urban areas.

The duration of hospital treatment was 3.0 ± 1.7 days. The majority of treated patients (96%) were adults (35 ± 15 years old), but surgical treatment was implemented just as frequently in the pediatric population as in the adult population (*χ^2^*, *p* = 1.0). A total of 347 patients (89.6%), including 14 children (15 ± 1.9 years old), were treated surgically by open rigid internal fixation (ORIF).

### 3.4. Mandibular Head Fractures

Next, an analysis was performed on only the 59 mandibular head fractures that are included in the clinical material (Figure 6). These are the fractures that are the most difficult to treat. In these fractures, surgical treatment was performed in the majority of patients (only four were cured by the closed method). The average length of hospitalization was 3.7 ± 1.7 days, which was significantly longer (*p* < 0.001) than for fracture treatment when there was no mandibular head fracture (3.0 ± 1.6 days). The fractures affected females less than males (14:45). Nine cases and 50 cases of mandibular head fractures were reported in rural and urban areas, respectively, in 5 children and 54 adults, as a result of assault (9 cases), vehicle accident (18 cases) and fall (30 cases) with no predilection for residence or sex.

Thirty-three cases of unilateral fractures and twenty-six cases of bilateral fractures occurred. There were 48 head fractures, 2 head fractures and concomitant high-neck fractures on the opposite side, 2 cases of head fracture and concomitant low-neck fracture on the opposite side, and 7 cases where base fractures occurred on the opposite side. The occurrence of bilateral fractures, when a component of at least one fracture is a head fracture, does not depend on the patient’s age, sex, place of residence, or cause of the fracture. They are also not associated with concomitant fractures of the body, angle, ramus, or coronoid process of the mandible.

Bilateral fractures (in which one is a fracture of the mandibular head, Figure 7) show combinations with both types of neck fractures contralaterally (*p* < 0.01). Concomitant fractures of type A with type B, type A with type C, and type B with type C on the opposite side also appeared (*p* < 0.01), while there were significantly more type B fractures in those with unilateral fractures than in those with bilateral fractures (*p* < 0.01). Bilateral fractures of the condylar process (in which one is a fracture of the mandibular head) are more often accompanied by additional fractures outside the condylar process (*p* < 0.05).

### 3.5. Medical Literature Interest in Mandibular Head Fractures

The authors retrieved 1194 publications on head fractures (*Fracture*) and 871 publications on their surgical treatment (*Surgical Treatment*), most of which, 827, were written in the last 40 years. It was found that the annual number of publications can be described by a regression equation. The number of publications focused on mandibular head fractures increased annually according to the equation (correlation coefficient, CC = 0.79; R^2^ = 63%; *p* < 0.001):*Fracture = −1581.12 + 0.000401549 ● Year of Publication^2^*(1)

The number of publications addressing the surgical treatment of mandibular head fractures increased annually according to the equation (CC = 0.85; R^2^ = 72%; *p* < 0.001):*Surgical Treatment = (−138.912 + 0.0000357036 ● Year of Publication^2^)^2^*(2)

The distribution of the annual number of publications on mandibular head fractures is different from the annual distribution of publications on the surgical treatment of these fractures. From the analysis of the increases presented in both kinds of publications from 1982 to 2022, the results show that the dynamics of the increase in the number of studies on the surgical treatment of mandibular head fractures is greater than the dynamics of the increase in the number of publications on mandibular head fractures (which is a constant increase). The query algorithm and search results are shown in Figure 8.

## 4. Discussion

The epidemiological data presented here indicate and confirm that trauma to the chin region causes fracture of the condylar process of the mandible in an indirect mechanism. A skin wound in the chin region is often seen in these patients. The epidemiology of fracture has been unchanging in recent decades: assaults [9] and falls [23] are the most common causes of condylar fracture. The causes of fracture occurrence in this study were similarly represented. There is certainly variation in the causes of injury by region and country. In general, falls are the most common cause of fractures worldwide. However, in Oceania and southern sub-Saharan Africa, assaults are the leading cause. The effects of armed conflict and terrorism in Ukraine, North Africa, and the Middle East can also be included in the same category of causes. Since the surgical treatment of fractures of the condylar process of the mandible, especially the mandibular head, can require considerable knowledge and the disability caused by these fractures can be significantly reduced by such treatment, it is important for healthcare systems across the world to develop programs with the aim of preventing injury and training specialists to treat fractures of the mandibular head. In addition, it is important to emphasize the need for more extensive data collection, better classification reflecting the actual epidemiology of fractures, and more thoughtful use of data to identify problems that can be addressed with current resources [1].

It is also worth noting the variability of regional fracture admissions. The geographic distribution of prevalent cases was also similar to that of incident cases. In 2017, the age-standardized prevalence of facial fractures was highest in Central Europe, with 68 cases per 100,000, representing 92 387 prevalent cases. Central Europe, Slovenia, and the Czech Republic had the highest age-standardized prevalence, with identical prevalence of 81 cases per 100,000, while Poland had the highest total number of prevalent cases, with 31,345 total cases in 2017 [1]. This probably partly explains the ease of gathering a group of patients with mandibular fractures and calculating the actual current incidence of mandibular head fractures.

In the last 20 years, advances in surgical techniques that are safe for cranial nerves have been documented [24,25,26,27,28] to ensure proper bone consolidation [29,30,31]. This became the basis for ORIF to become the widespread treatment for patients diagnosed as having mandibular head fractures.

Decades ago, the incidence of mandibular head fractures had not yet been reported [2,3,4,5,6,7,32], while the frequency of diagnoses of all types of mandibular condylar process fractures was almost twice as low as it is today [32,33,34]. Sometimes a surprisingly low incidence was reported, e.g., 16.5% among all mandibular fractures [35]. The situation began to change a few years ago [23,36] in reports of diagnosis in children (56% of all mandibular fractures) [23]. The same thing was noted more than 40 years ago [9], although Lindahl recorded only 28 mandibular head fractures in a group of 123 patients included due to condylar fracture (23%). Forty-five years later [23], 41% of mandibular head fractures are diagnosed among all recognized fractures of the condylar process of the mandible. However, Mahgoub et al. collected only 27.5% of head fractures among mandibular condylar process fractures [36]. This is a large discrepancy, and the question remains: how common are mandibular head fractures currently?

In the group of patients included in the study, 45% suffered a mandibular condyle fracture. Among these fractures of the condylar process, the most common was a fracture of its base (54%), but the second most common was a fracture of the mandibular head (34%). Thus, of the total 175 fractures of the condylar process, as many as 59 involved an anatomical region that is difficult to treat, i.e., the mandibular head. Thus, fractures of the mandibular head are not occasional incidents but a very common posttraumatic pathology requiring highly specialized treatment. Thus, it seems that in light of the study of clinical material, the incidence of mandibular head fractures is closer to that reported by Shi et al. [23] and higher than that reported by Mahgoub et al. [36].

It should be noted that, in general, unilateral condylar fractures outnumber bilateral fractures by three times (75:25). However, among mandibular head fractures only, the proportions were almost equal (unilateral:bilateral = 33:26). They most often occur bilaterally as mandibular head fractures and less often as a combination of a head fracture and another fracture of the condylar process of the mandible (Figure 7). Undoubtedly, the collected results are influenced by the location of the department in an adult patient’s hospital. Pediatric multisite injuries, which include a mandibular fracture, will initially be treated in a pediatric trauma center. Several times a year, the surgical team from our center operates at the pediatric traumatology center. This is based on call collaboration and does not directly participate in diagnosis. Due to the lack of oral and maxillofacial surgery specialists on-site at the pediatric center, it is possible to underestimate mandibular condylar process fractures, especially mandibular head fractures.

Mandibular head fractures (Table 3) were statistically related (or strongly related, *p* < 0.001 for mandibular body or angle fractures) to any other mandibular fracture (*p* < 0.05). Therefore, especially in cases of diagnosed mandibular body and angle fractures, it is important to look for a second fracture fissure within the mandibular head. In addition, in the case of an angle fracture, another fracture of the base of the condyle or neck of the mandible should be expected. This is helpful in daily work with traumatology patients. If a patient with a laceration of the chin region reports a bicycle/scooter accident, a condylar fracture should be suspected (with high probability, *p* < 0.001) and should be sought for in further diagnostic procedures. In addition, if the patient has any fracture of the mandible and the two previously mentioned events, it is almost certain that this person has an additional fracture of the condylar process of the mandible. It should be remembered that isolated fractures of the mandibular condylar process alone, without any other fracture, are quite rare, i.e., in 17% of mandibular fractures. It should also be emphasized that fractures of the mandibular head occur twice as often in the pediatric population as in adults. To limit exposure to ionizing radiation, mandibular fractures can be easily diagnosed by orthopantomogram [37,38,39]. However, in a pediatric patient after a traffic accident with a chin injury, a CT scan with proper imaging of the mandibular heads should be used to exclude such a fracture.

It seems that in a medical center providing surgical treatment for mandibular head fractures, there is a greater possibility of making a diagnosis of mandibular head fracture. However, there is no overinterpretation because later, these patients are treated in a medical center providing surgical treatment for mandibular head fractures. In contrast, at medical centers providing closed or surgical treatment of lower mandibular condylar process fractures, underdiagnosis can be expected. Misidentification, inadequate treatment, or lack of treatment can lead to permanent aesthetic deformity or functional issues [32].

Assessing the number of publications on head fractures of the mandible and their surgical treatment can be an additional point in understanding the increasing number of recognized head fractures. However, it should be said that this is a very narrow specialty (only 827 publications in the past 40 years). It is possible to notice an increase in interest in the surgical treatment of such fractures after 1998, i.e., since the publication of the milestone study by Kremer et al. [40]. Publications related to the surgical treatment of mandibular head fractures increased rapidly in the 2000s. Kermer, Undt, and Rasse established the current standard for the surgical treatment of mandibular head fractures with the application of long screws. Since then, new screws have been invented [11,41], including compression headless screws [42,43,44,45,46,47] and metallic biodegradable screws [48,49,50,51]. Currently, this technique has been developed with very sophisticated computer-assisted capabilities [52,53,54]. From 1998 to 2022, there were 711 publications out of a total of 871 published. In regression analysis of the number of annual publications on fractures and publications on surgical treatment, one can see an increasing trend in the number of publications on the surgical treatment of mandibular head fractures. The authors attribute this trend, and the widespread availability of computer tomography scanners [15], to the increasing number of diagnoses of mandibular head fractures in epidemiological descriptions, in addition to a known surgical technique [55] to ensure proper consolidation [29] of anatomically reduced mandibular head fragments (which increases surgeons’ willingness to ORIF these technically difficult fractures): See Table 4.

However, in general, the annual number of scientific publications is increasing by 4%, with the simultaneous forgetting of old, significant, and still current scientific texts. This phenomenon is a certain limitation of the literature review presented here. Note, however, that groundbreaking articles from more than 50 years ago are cited here [2,3,4,5,6,7,9,13,16,38,56,57]. It is concerning that citations are rising by 5.6% per year, with them doubling every 12 years among scientific papers with a gap in the citation of brand new or very old research [58]; the strength of this paper is that there are also many citations of research that is younger than six years old [1,14,17,18,23,24,28,29,30,34,36,42,43,44,45,46,48,49,50,51,52,53,54,56].

## 5. Conclusions

Mandibular head fractures are not as rare as previously thought. They represent more than a third of all condylar process fractures.

A child after a head injury is a suffering and a difficult patient who often cannot accurately describe the perceived functional disturbances that a mandibular head fracture can cause. Knowing that mandibular head fracture occurs twice as often in the pediatric population as in the adult population, appropriate emphasis should be placed on the complete diagnosis of temporomandibular joint conditions in the pediatric trauma patient. With the assistance of computer tomography scanners, a correct diagnosis is readily achievable.

It is worth realizing that a mandibular fracture is most likely related to a fracture of the head of the mandible. This observance can guide the diagnostic procedure in the future and allow early, single-stage treatment for every fracture under a single anesthesia induction.

## Figures and Tables

**Figure 1 jcm-12-01394-f001:**
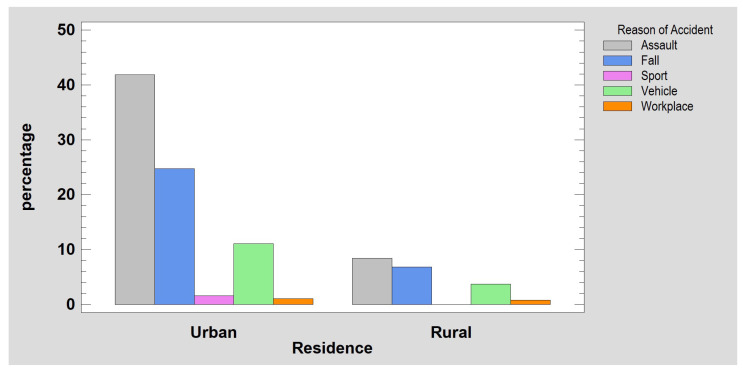
Frequency of observed mandibular fractures in urban and rural populations (no significant differences).

**Figure 2 jcm-12-01394-f002:**
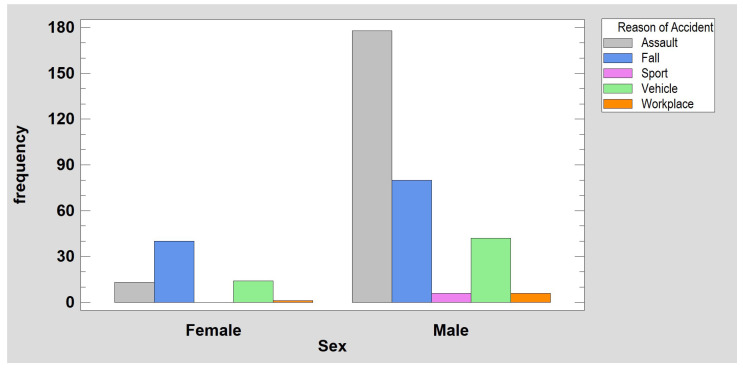
Distribution of causes of mandibular fractures by sex. Women had significantly more fractures due to falls, and men had significantly more fractures due to assaults (*p* < 0.001).

**Figure 3 jcm-12-01394-f003:**
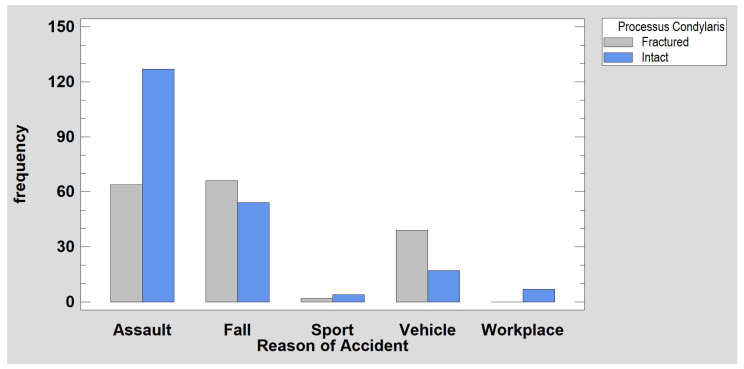
The relationship between the cause of injury and the occurrence of condylar fracture in the mandible. Note that fractures resulting from falls and vehicular accidents are more common (*p* < 0.001) than the “intact” status of this anatomical region (i.e., they cause fractures of other parts of the mandible).

**Figure 4 jcm-12-01394-f004:**
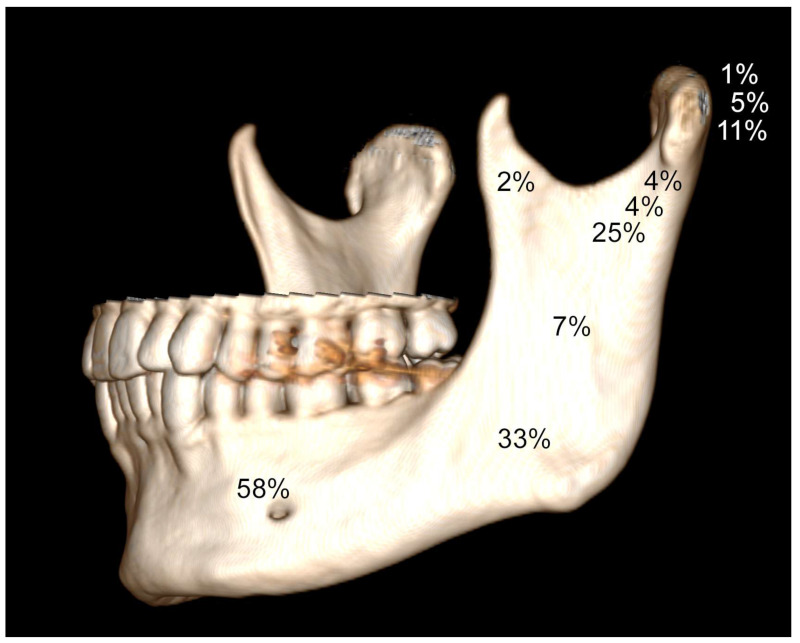
Frequencies of observed fractures of the mandible (as presented in Table 2). Data do not sum to 100% because there are multiple fractures in the included cohort.

**Figure 5 jcm-12-01394-f005:**
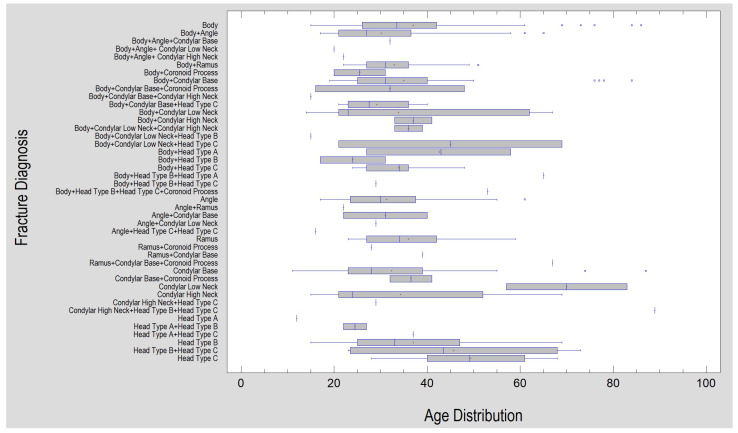
Fracture patterns observed in the presented material. In this overview, they refer to the age of the treated patients, but it is also important to note the rarity of certain fracture combinations (vertical lonely lines on the graph), e.g., fractures of the condylar process neck rarely co-occur with fractures of the mandibular body and angle. It is worth noting this and always remembering the importance of looking for another fracture of the condylar process, even when it seems that such a mandible fracture has never been observed together with a fracture of the condylar process of the mandible. Eighteen such peculiar combinations were recorded.

**Figure 6 jcm-12-01394-f006:**
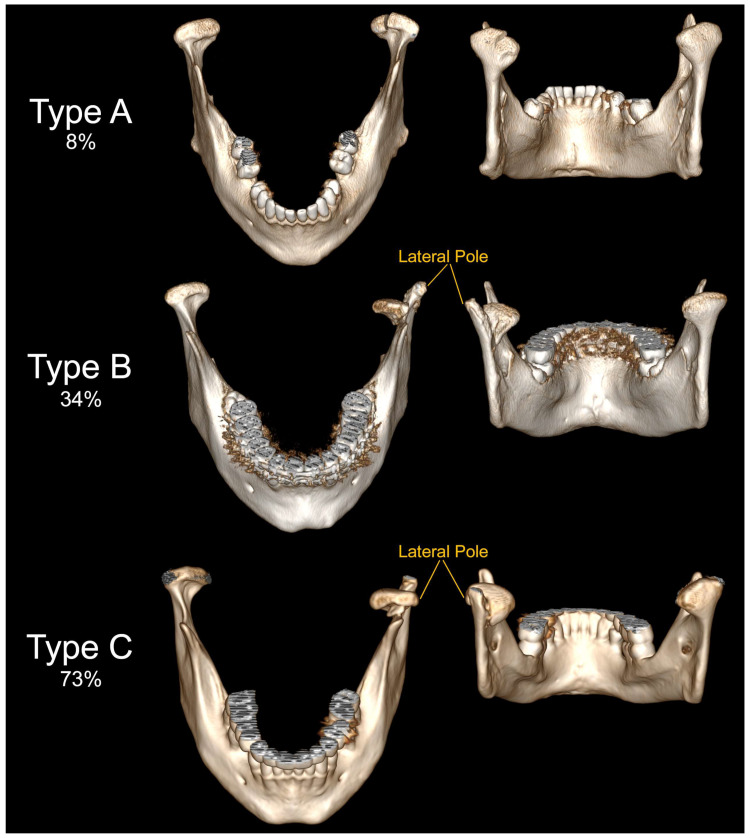
Examples of fractures of the mandibular head observed in the collected clinical material. The fracture occurs in the picture above within the left condylar process. Classification according to Neff: Type A—the fracture line runs through the articular cartilage close to the medial pole of the mandibular head (observed in 8% of mandibular head fractures); Type B—the fracture line runs just at the medial pole of the mandibular head (in this case, the lateral pole is not damaged) (observed in 34% of mandibular head fractures); Type C—the fracture line runs just below the lateral pole of the mandibular head (the lateral pole is detached to the distal fragment and is displaced with the entire mandibular head forward and downward) (observed in 73% of mandibular head fractures).

**Figure 7 jcm-12-01394-f007:**
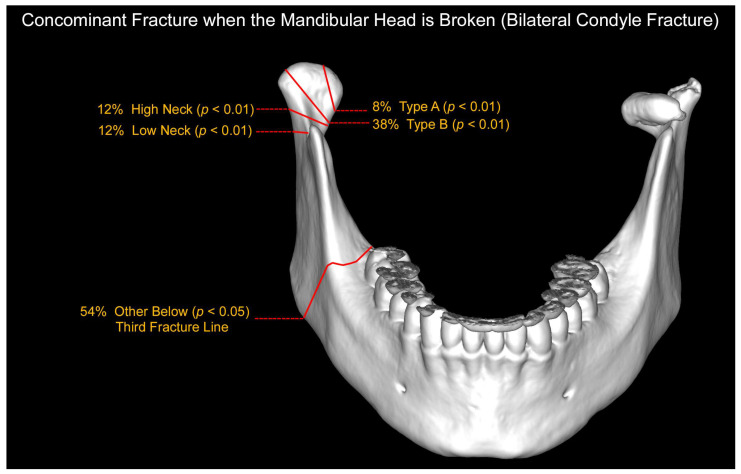
Evaluation of the location of a concomitant fracture in patients with bilateral mandibular condylar process fractures when one of the fractures is a mandibular head fracture. Most often, the second mandibular condylar process fracture is a type B head fracture, followed by neck fractures and a type A head fracture. At the same time, more than half of the patients have a third and further fracture fissure outside the condylar processes (most often a fracture in the mandibular body).

**Figure 8 jcm-12-01394-f008:**
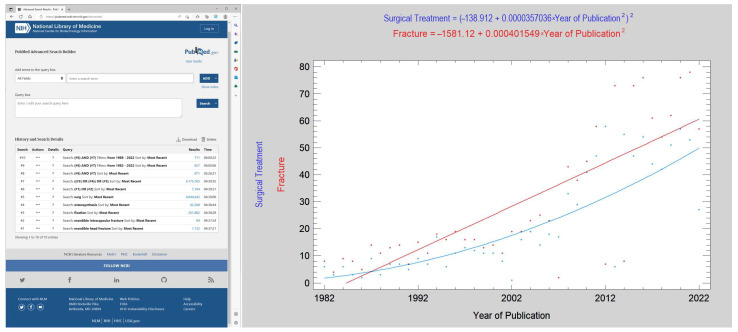
Annual number of publications on fractures of the mandibular head (Fracture) and surgical treatment of fractures of the mandibular head (Surgical Treatment). An attempt to explain the increasing incidence of mandibular head fracture diagnosis. The panel on the left shows how publications on mandibular head fractures and surgical treatment were sought. It seems that the possibilities of surgical treatment of these fractures have a greater ability to diagnose them.

**Table 1 jcm-12-01394-t001:** Detailed reasons for mandibular fractures (%, *χ^2^* test of independence).

Reason of Fracture	Residence		Sex		Age		Location	
	Urban	Rural	Female	Male	Pediatric	Adult	Condylar	Rest of Mandible
Assault	42.11	8.68	3.68	47.11	0.79	50.00	16.84	33.95
Car	3.16	0.53	0.79	2.89	0.26	3.42	1.84	1.84
Motorcycle	0.00	0.26	0.00	0.26	0.00	0.26	0.00	0.26
Bike/Scooter	7.89	2.89	2.89	7.89	1.58	9.21	8.42	2.37
Fall on the Ground	20.26	5.53	9.21	16.58	0.79	25.00	14.21	11.58
Fall on the Stairs	1.05	0.26	0.53	0.79	0.00	1.32	0.79	0.53
Fall from a Height	3.42	1.05	0.79	3.68	0.79	3.68	2.37	2.11
Sports	1.84	0.00	0.00	1.84	0.00	1.05	0.53	1.32
Workplace	0.53	0.53	0.00	1.05	0.00	1.05	0.00	1.05
Total	80.26	19.74	17.89	82.11	4.21	95.79	45.00	55.00
Statistical Difference	No significant		*p* < 0.001		*p* < 0.01		*p* < 0.001	

**Table 2 jcm-12-01394-t002:** Frequency of mandible fractures.

Fracture Site	Count	Percentage
Body	255	58% of all fractures
Angle	127	32% of all fractures
Ramus	26	7% of all fractures
Coronoid Process	9	2% of all fractures
Condyle Base	96	25% of all fractures
Condyle Low Neck	16	4% of all fractures
Condyle High Neck	16	4% of all fractures
Mandible Head C	43	11% of all fractures
Mandible Head B	20	5% of all fractures
Mandible Head A	5	1% of all fractures
Mandible Neck	28	7% of all fractures
Mandible Head	59	15% of all fractures
Condylar Process	175	45% of all fractures
Condyle Base	96	54% of condyle fractures
Condyle Low Neck	16	9% of condyle fractures
Condyle High Neck	16	9% of condyle fractures
Mandible Head C	43	3% of condyle fractures
Mandible Head B	20	11% of condyle fractures
Mandible Head A	5	25% of condyle fractures
Neck	28	16% of condyle fractures
Head	59	34% of condyle fractures
Type C	43	73% of head fractures
Type B	20	34% of head fractures
Type A	5	8% of head fractures

**Table 3 jcm-12-01394-t003:** Analysis of multiple fractures. Interrelationship of fracture sites in mandible.

Angle	Ramus	Condyle Base	Neck	Head *	Fracture Location
*p* < 0.05	Not significant	Not significant	Not significant	*p* < 0.001	**Body**
	*p* < 0.05	*p* < 0.001	*p* < 0.05	*p* < 0.001	**Angle**
		*p* < 0.05	Not significant	*p* < 0.05	**Ramus**
			*p* < 0.05	*p* < 0.05	**Condyle Base**
				*p* < 0.05	**Neck**

* within head fractures, type B coexists with type A on the opposite side (*p* < 0.001) or with type C (*p* < 0.001); no relation was found between fractures type A and type C.

**Table 4 jcm-12-01394-t004:** Factors contributing to the real proportion of mandibular head fractures in published studies.

N	Factor
1	Availability of CT scanners
2	Development of biomechanical knowledge of the mandible
3	New methods of osteosynthesis
4	Development of fixation materials (screws)
5	New surgical instruments
6	Widespread access to highly specialized knowledge
7	Long-term observation of the results of closed and surgical treatment
8	Increasing frequency of mandibular head fractures in epidemiological publications *

* positive feedback mechanism.

## Data Availability

The data on which this study is based will be made available upon request at https://www.researchgate.net/profile/Marcin-Kozakiewicz (accessed on 1 January 2023).

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
