# Peer review of "Current Frequency of Mandibular Condylar Process Fractures"

_jcm, 2023, doi:10.3390/jcm12041394_

Round 1
Reviewer 1 Report
The papier has a high clinical value and implications. I would like to ask the authors to further comment the sentence from Conclusions section:Head fractures occur twice as often in the pedriatic population than in adults.
Authors should add: What is the main question addressed by the research?
Author Response
Please, find the attached letter

Reviewer 2 Report
Dear authors !
I was given the opportunity to review Your manuscript investigating the current frequency of mandibular condyle fractures.
First, it is totally unclear if the authors present a retrospective analysis of a patient sample from an OMF-surgery department or a systematic review of the topic.
Second, the authors inappropriately present results in the Material and Methods section.
Third, the presentation of the results is confusing since the targets of the research are not defined at all (there is no description of the primary outcome evaluation, possible second, third etc. outcome evaluations).
The entire manuscript is missing a clear structure and by this is incomprehensible, the results are presented without comprehensible structure.
Last but not least, it is not written in comprehensible English language.
Thank You
Author Response
Please, find the attached letter.

Reviewer 3 Report
This is a well written article on the frequency of mandibular fractures. I have a few comments to further improve the paper.
Sex and age differences in Table 1 should also be included in the text. I believe that age (older) and sex (female) are also predisposing factors of mandibular fracture. Some fractures are also more common in older people as seen in Figure 5, particularly the “condylar low neck”
A total of 1194 publications on head fracture and 871 publications on surgical treatment were retrieved. I believe that these numbers are straight from the search results. First, the authors mention they were articles on “head fracture” Do you mean mandibular head fracture? The term “head fracture” should be replaced with “mandible head” or “mandibular head” throughout the paper to avoid confusion. Also, was each individual entry evaluated? How can one be sure that these publications were about mandibular fracture.
The increase in the number of publications over the years might simply reflect the growth of scientific publications/journals/publishers in recent years. Here is one example: https://www.nature.com/nature-index/news-blog/the-growth-of-papers-is-crowding-out-old-classics. This should be discussed and noted as a limitation.
The authors should discuss the presence of Stafne defect, a mandibular defect which increases the likelihood of mandible fracture (https://doi.org/10.1016/j.joms.2009.06.019). This defect is present in 0.17% according to a recent meta-analysis (https://doi.org/10.1016/j.jds.2022.08.022).
Author Response
Please, find the attached letter.
